# Witchweed's Suicidal Germination: Can Slenderleaf Help?

**Fridah A. Mwakha [1], Nancy L.M. Budambula [2], Johnstone O. Neondo [3], Bernard M. Gichimu [1], Eddy O. Odari [4], Peter K. Kamau [5], Calvins Odero [6], Willy Kibet [6] and Steven Runo [6,\*]**

1. Department of Agricultural Resource Management, University of Embu, P.O. Box 6, Embu 60100, Kenya; fridahmwakha@gmail.com (F.A.M.); wacikubm@gmail.com (B.M.G.)
2. Department of Biological Sciences, University of Embu, P.O. Box 6, Embu 60100, Kenya; budambula.nancy@embuni.ac.ke
3. Institute of Biotechnology Research, Jomo Kenyatta University of Agriculture and Technology, P.O. Box 62000, Nairobi 00200, Kenya; jneondo@jkuat.ac.ke
4. Department of Medical Microbiology, Jomo Kenyatta University of Agriculture and Technology, P.O. Box 62000, Nairobi 00200, Kenya; kodarie04@yahoo.com
5. Department of Life Sciences, South Eastern Kenya University, P.O. Box 170, Kitui 90200, Kenya; pkuria@seku.ac.ke
6. Department of Biochemistry, Microbiology and Biotechnology, Kenyatta University, P.O. Box 43844, Nairobi 00100, Kenya; calvins.odero@gmail.com (C.O.); kibetwilly35@gmail.com (W.K.)
* Correspondence: runo.steve@ku.ac.ke

**Abstract:** The parasitic plant *Striga hermonthica* (Delile) Benth. is stimulated to germinate by biomolecules (strigolactones) produced in the roots of host and some non-host plants. Non-hosts induce *Striga*'s suicidal germination and are therefore used as trap crops. Among trap crops, the Slenderleaf legume in the genus *Crotalaria* (*Crotalaria brevidens* (L.) Benth.) and (*Crotalaria orchroleuca* (G.) Don.) has been popularized in African smallholder farms. However, the *Striga* germination efficiency of these locally grown *Crotalaria* varieties (landraces) is unknown. Also unclear is *Crotolaria*'s extent to inhibiting *Striga* growth, post germination. Extensive parasite penetration can expose the trap crop to secondary infections and possible phytotoxicity from *Striga*. We used *in vitro* germination assays to determine the *Striga* germination efficiency of 29 *Crotalaria* landraces. Furthermore, we determined *Crotalaria*'s ability to inhibit *Striga* attachment and growth using histological analysis. We found that: (i) *Crotalaria* stimulated germination of *Striga* seeds at frequencies ranging between 15.5% and 54.5% compared to 74.2% stimulation by the synthetic strigolactone (GR24) used a positive control; (ii) *Crotalaria* blocked *Striga* entry at multiple levels and did not allow growth beyond the pericycle, effectively blocking vascular connection with the non-host. Hence, *Crotalaria* is suitable as a trap crop in integrated *Striga* management.

**Keywords:** *Crotalaria*; *Striga*; strigolactones; suicidal germination; trap crops

## 1. Introduction

The *Striga* genus belong to Orobanchaceae family which comprise more than 2000 species of parasitic plants that parasitize major tropical cereals and some legumes [1]. Of the many *Striga* species, (*Striga hermonthica* (Delile) Benth.) and (*Striga asiatica* (L.) Kuntze) are the most widespread and infect all cereals including maize, sorghum, rice, and millet. Other *Striga* species of economic importance are (*Striga gesnerioides* (Wild.) Vatke) which attacks cowpea (*Vigna angucuilata* L. Walp.) and peanut (*Arachis hypogea* L.) [1].

*Striga* causes yield losses of between 10–100%, leading to economic losses of about USD 7 billion every year [2]. The infestation is most severe in the majority of African smallholder farms where mono-cropping and continuous cultivation of *Striga* host crops is practiced under nutrient depleted soils. Ineffective control methods further exacerbate the problem because of deposition of a large amount of seeds in soil (*Striga* seedbank) [3]. These seeds maintain dormancy and are viable for decades until stimulated to germinate by a suitable plant [4] which, produce germination stimulants called strigolactones [5]. Taking this into account, meaningful *Striga* control can only be achieved using an integrated approach that leads to significant reduction of the *Striga* seedbank in soil.

*Striga* seedbank depletion can be achieved by suicidal germination – a strategy where the *Striga* seeds are stimulated to germinate in absence of a true-host plant [6]. Chemical stimulants of suicidal germination have been identified and they include synthetic analogues of strigolactones such as MP1, MP3 and Nijmegen1 [7]. However, chemical synthesis of strigolactones is expensive and beyond the reach of smallholder farmers in Africa. In these regions, legumes have been identified as attractive trap crops because of their ability to stimulate *Striga* germination while improving soil fertility [8]. Among these, the cattle forage legumes, Silverleaf Desmodium (*Desmodium uncinatum* (Jacq.) DC) and Greenleaf Desmodium (*D. infortum* (Mill.) Urb.) have been highly exploited in reduction of the *Striga* seedbank in smallholder farms [9]. However, Desmodium is not a food crop and this has limited its adoption in most households [10]. As an alternative, food legumes have been advocated as attractive methods for controlling *Striga* infestations because similar to *Desmodium spp.*, they too can stimulate seeds of *S. hermonthica* to germinate [11]. Food legumes also accrue additional benefits as sources of food, nutrition, and improved livelihoods.

In this study, we sought to determine suitability of the legumes (*Crotalaria brevidens* (L.) Benth.) and (*Crotalaria orchroleuca* (G.) Don.) as trap crops for *S. hermonthica*. These legumes are indigenous vegetables widely consumed in Kenya and other African countries as sources of proteins, zinc, iron, and vitamin [12]. They also improve soil fertility when used as green manure due to high biomass production and their natural ability to fix atmospheric nitrogen [13]. In fact, *Crotalaria* has been shown to reduce *Striga* infestation and also increase yields under field conditions [8]. However, literature on effectiveness of different *Crotalaria* species and varieties to induce germination of *S. hermonthica* is not available. Furthermore, it is not known the extent to which *Crotalaria* is able to prevent *Striga* attachment. Non-host incompatibility in *Striga* can be at different levels such as cortex, pericycle, or endodermis [14]. In some cases, there can be vascular connection between a *Striga* and a non-host [14]. The level of incompatibility is important because *Striga* infections may subject the trap crop to secondary infections. It is also possible that during the the initial stages of parasitism, *Striga* could adversely affect a host or non-host through phytotoxicity. Support for such a hypothesis can be drawn from recent work that showed toxicity of rice (*Oryza sativa* L.) tissue that was co-cultivated with *Striga* under *in vitro* conditions [15]. The phytotoxity is now believed to be – at least in part – due to abscisic acid [16]. Taking these into account, a trap crop should stimulate a large number of *Striga* seeds to germinate and have minimal interaction with the parasite.

We therefore tested the hypothesis that locally grown cultivars of *Crotalaria*—described as landraces in this study exhibit variations in their ability to stimulate suicidal germination of *S. hermonthica* seeds. We also sought to determine the extent of interaction between *Striga* and the non- host *Crotalaria*.

## 2. Materials and Methods

### 2.1. Plant Materials

Seeds of 13 landraces of *C. brevidens* and sixteen of *C. ochroleuca* were collected from farmer's seed stock (collections of *Crotalaria* seeds propagated and maintained by farmers) in nine western Kenya Counties. Global positioning system (GPS) coordinates for collection sites as well as landrace information are presented in Table S1. Mature *Striga* seed heads were collected from sorghum fields in western Kenya (Kisumu; 0.0699°S, 34.8169°E) during the 2015 growing season and put in paper

bags. Subsequently, heads were threshed by tapping the papers and seeds separated from debris by passing through 250 and 150-micron sieves using methods described by [17]. Cleaned *Striga* seeds were maintained in opaque glass containers containing silica gel packets until use.

## 2.2. Preparation of Crotalaria Root Exudate

*Crotalaria* (Slenderleaf) seeds were pre-germinated in plastic planters filled with vermiculite for a period of 14 days in triplicates arranged in completely randomized design (CRD). Seedlings were allowed to grow until they produced the first trifoliate leaves and well-developed transferable roots. Fifteen seedlings were used per landrace. Each seedling was transferred into 40 mL glass test tubes containing long Ashton nutrient solution [18] and anchored upright using cotton wool. Tubes were then wrapped with aluminum foil and left in the glasshouse for seven days after which seedlings were washed and the nutrient solution replaced with sterile distilled water in 20 mL test tubes. The root exudate was then collected after 48 h.

## 2.3. Striga Germination Bioassays

Prior to germination, *Striga* seeds were preconditioned according to the protocol of Mbuvi et al. [19]. Twenty-five (25) mg (approximately 5000 seeds) were surface sterilized in 25 mL of 10% (*v/v*) Sodium hypochlorite (commercial bleach) for 10 min with gentle agitation. The seeds were later emptied on a funnel lined with a Whitman GFA filter paper and rinsed with 250 mL of sterile distilled water [18]. Seeds on the filter paper were then placed on 9 cm diameter petri dishes and 5 mL of sterile distilled water added. Petri plates were then sealed with parafilm, wrapped in aluminum foil and incubated at 29 °C for 11 days [18].

Germination assays using *Crotalaria* were carried out by adding 5 mL of the crude root exudate to petri dishes containing preconditioned *S. hermonthica* seeds. For the positive and negative controls 5 mL of 0.1 ppm GR24 (Chiralix, Amsterdam (http://www.chiralix.com/rightclick.cfm?id=67359)) and distilled water were added. All the treatments were replicated three times in a CRD design and then incubated for 24 h in an incubator set at 30 °C.

Petri dishes with germinating *S. hermonthica* seeds were observed and photographed under Leica model MZ7F stereomicroscope (Leica, Germany) fitted with a DFC320FX camera (Leica, Germany). Seeds were considered to have germinated when a white radicle protruded through the seed coat. Photographs were analysed using ImageJ v. 1.45 (http://rsb.info.nih.gov/ij) to determine the radicle length and the seed germination frequency. Analysis of the photographs using imageJ software was done by first setting a scale in mm and measuring the seed and radicle length using the curve feature of ImageJ. Germination frequency was calculated as a percentage of germinated seeds relative to the total number of seeds. Recorded data were first subjected to two-tailed Student t-test to determine significance differences in germination frequencies and radicle lengths between *C. brevidens* and *C. orchroleuca,* followed by one-way analysis of variance (ANOVA) using statistical analysis software (SAS) version 9.4. Their corresponding means were separated using Student Newman Keul's (SNK) test at 95% confidence interval. Pearson correlation analysis was done between the radicle length and germination frequency using XLSTAT version 2019.

## 2.4. Post germination Analysis of Striga-Crotalaria Interactions

To determine levels of incompatibility between *Striga* and *Crotalaria*, seeds of landraces inducing the lowest, medium, and highest *Striga* germination frequencies (MKSM0218, MKSM0204, and BMGR0234) were pre-germinated as described in Section 2.2 above. A susceptible local maize landrace (Namba Nane) was also included as a check for compatible *Striga*-host interaction. Germinated seedlings were then transferred to rhizotrons – soil free root observation chambers prepared as described in Mbuvi et al., [19]. Rhizotrons containing *Crotalaria* and maize seedlings were maintained in the glasshouse for 21 days while supplied with 25 mL of 40% Long Ashton nutrient solution [18] daily.

After 21 days, *Crotalaria* and maize seedlings with well-developed roots were infected with 25 mg of pre-germinated *S. hermonthica* seeds by inoculating them using a soft paintbrush. Five *Crotalaria* and maize plants were screened and the experiment replicated three times. Close-up photographs showing interactions of *Striga-Crotalaria* and *Striga*-maize were taken at 9 days after infection (DAI) using a Leica MZ10F stereomicroscope fitted with a DFC310FX camera (Leica, Germany).

Histological analysis of parasite development within the *Crotalaria* or maize roots were carried out on the landrace MKS21 because it supported *Striga* growth beyond the non-host's cortex. For comparison, histological analysis was also carried out on *Striga*-maize interaction using the landrace Namba Nane. Protocols for histological analysis were as described in Mutinda et al., [20]. Briefly, small tissues with *Striga* haustoria attachment were dissected from roots at 9 DAI. Samples were fixed using Carnoy's fixative – 4:1, ethanol: acetic acid [21] then dehydrated in 100% ethanol for 30 min. Tissues were then stained with 1% Safranin O in 30% ethanol for 5 min and de-stained using choral hydrate (2.5 g/mL) for 12 h before being photographed.

For embedding, we used the Technovit$^{®}$ 7100 kit (Haraeus Kulzer GmbH. Samples were pre-infiltrated in ethanol-Technovit 1) solution (1:1) for one hour then infiltrated in 100% Technovit 1 solution for 15 min. Fresh Technovit 1 solution was added to the samples and maintained at room temperature for 3 days to complete the infiltration process. Tissues were embedded vertically in a preparation of Hardener 2 and Technovit 1 solution, 1:15 ratio respectively, contained in 1.5 mL micro-centrifuge lids. The preparation was left in open air to dry for a week before wrapping them in aluminum foil for incubation at 37 °C to facilitate drying. Moulds were mounted onto wooden blocks using the Technovit$^{®}$ 3040 kit following the manufacturer's instructions (Haraeus Kulzer GmbH). For sectioning, 5 micron-thick slices were cut using a Leica RM 2155 microtome (Leica instruments) then transferred to microscope slides. Sections were stained using 0.1% toluidine blue O (Sigma, St. Louis, MO, USA) in 100 mM phosphate buffer at pH 7 for two minutes, then washed in distilled water and dried at 65 °C for 30 min on a hot plate. Slides with sections were then covered with slips using DePex (BDH, Poole, UK), observed, and photographed using a Leica DM100 microscope (Leica, Germany) fitted with a Leica MC190HD camera (Leica, Germany).

## 3. Results

### 3.1. Crotalaria Root Exudates Stimulate Germination of S. hermonthica Seeds

Germination frequencies of *Striga* induced by the 29 *Crotalaria* landraces varied significantly at ($p < 0.0001$). There were no significant differences in germination frequency between *C. brevidens* and *C. orchroleuca* species as revealed by t-test with a *p* value of 0.5970 and t-value of −0.5307. The germination frequency means of landraces ranged from 15.5% to 54.5%. Distilled water (negative control) had no observable germination induction effect while the positive control GR24 had the highest germination frequency of 76.1% that was significantly different from the other treatments (Table 1; Figure 1). *C. brevidens* landrace from Homabay coded as BHM213 had the highest germination frequency of 54.5%. It was significantly different from the other landraces except a *C. brevidens* landrace from Migori (BMG234) which recorded the second highest germination frequency of 51.94%. The rest of *C. brevidens* landraces from Siaya, Kakamega, Vihiga, Kisumu recorded lower germination frequencies ranging from 17.9% to 32.5%. The *C. ochroleuca* landrace from Kisumu coded as MKM218 induced the lowest germination frequency (15.5%). Most of the *C. ochroleuca* landraces from Kisumu, Homabay, Kakamega, Migori and Bungoma induced moderate germination (30.1–44.9%).

**Table 1.** Stimulation of *Striga* seed germination by *Crotolaria* determined using germination frequencies and radicle lengths of germinated *Striga* seedlings.

| S/NO | Landrace | Species | Germination Frequency (%) | Radicle Length (mm) |
|------|----------|---------|---------------------------|---------------------|
| 1 | GR24 | Synthetic Analog | 76.2 ± 4.20 [a] | 0.55 ± 0.06 [bcd] |
| 2 | BHMBY0213 | *Crotalaria brevidens* | 54.6 ± 3.01 [b] | 0.21 ± 0.02 [jk] |
| 3 | BMGR0234 | *Crotalaria brevidens* | 51.9 ± 0.40 [b] | 0.50 ± 0.05 [bcdef] |
| 4 | MKKMG0011 | *Crotalaria ochroleuca* | 44.9 ± 1.31 [c] | 0.43 ± 0.00 [cdefg] |
| 5 | MKS0226 | *Crotalaria ochroleuca* | 37.5 ± 1.13 [d] | 0.35 ± 0.02 [bfghij] |
| 6 | MVG0003 | *Crotalaria ochroleuca* | 36.7 ± 1.52 [de] | 0.42 ± 0.04 [cdefgh] |
| 7 | MMGSR0029 | *Crotalaria ochroleuca* | 36.5 ± 1.52 [de] | 0.24 ± 0.02 [ijk] |
| 8 | MKSM0204 | *Crotalaria ochroleuca* | 35.2 ± 0.06 [def] | 0.78 ± 0.08 [a] |
| 9 | MSY0110 | *Crotalaria ochroleuca* | 34.8 ± 1.69 [defg] | 0.39 ± 0.04 [defghi] |
| 10 | MHMBY0207 | *Crotalaria ochroleuca* | 34.2 ± 1.87 [defg] | 0.24 ± 0.03 [ijk] |
| 11 | MKS0221 | *Crotalaria ochroleuca* | 33.75 ± 1.75 [cde] | 0.41 ± 0.04 [cdefjhi] |
| 12 | BKSM0203 | *Crotalaria brevidens* | 32.5 ± 1.28 [defgh] | 0.16 ± 0.03 [kl] |
| 13 | MMGR0234 | *Crotalaria ochroleuca* | 32.1 ± 0.79 [defgh] | 0.61 ± 0.01 [b] |
| 14 | MKKMG0111 | *Crotalaria ochroleuca* | 30.8 ± 1.08 [efghi] | 0.72 ± 0.01 [a] |
| 15 | MBG0086 | *Crotalaria ochroleuca* | 30.1 ± 0.52 [fghij] | 0.43 ± 0.02 [cdefgh] |
| 16 | MHMBY0215 | *Crotalaria ochroleuca* | 28.9 ± 0.64 [fghij] | 0.56 ± 0.03 [bc] |
| 17 | BMGR0239 | *Crotalaria brevidens* | 28.6 ± 0.21 [ghijk] | 0.51 ± 0.01 [bcde] |
| 18 | BSY0113 | *Crotalaria brevidens* | 27.1 ± 0.11 [hijk] | 0.78 ± 0.02 [a] |
| 19 | BVG0001 | *Crotalaria brevidens* | 25.4 ± 1.02 [ijkl] | 0.25 ± 0.10 [hijk] |
| 20 | BHMBY0198 | *Crotalaria brevidens* | 24.7 ± 1.75 [ijklm] | 0.14 ± 0.0⁴ [kl] |
| 21 | BKSM0197 | *Crotalaria brevidens* | 23.7 ± 1.52 [jklmn] | 0.15 ± 0.03 [kl] |
| 22 | BKS0220 | *Crotalaria brevidens* | 23.7 ± 1.52 [jklmne] | 0.41 ± 0.03 [cdefghi] |
| 23 | BKKMG0091 | *Crotalaria brevidens* | 22.4 ± 0.52 [klmn] | 0.06 ± 0.00 [l] |
| 24 | MBS0065 | *Crotalaria ochroleuca* | 22.3 ± 0.71 [klmn] | 0.16 ± 0.00 [kl] |
| 25 | MVG0125 | *Crotalaria ochroleuca* | 22.0 ± 1.37 [klmn] | 0.27 ± 0.06 [ghijk] |
| 26 | MSY0216 | *Crotalaria ochroleuca* | 19.8 ± 1.11 [lmno] | 0.28 ± 0.01 [ghijk] |
| 27 | BVG0004 | *Crotalaria brevidens* | 19.8 ± 0.71 [lmno] | 0.36 ± 0.00 [efghi] |
| 28 | MBS0064 | *Crotalaria ochroleuca* | 18.4 ± 1.79 [mno] | 0.05 ± 0.00 [l] |
| 29 | BKKMG0129 | *Crotalaria brevidens* | 17.9 ± 0.05 [no] | 0.28 ± 0.00 [ghijk] |
| 30 | MKSM0218 | *Crotalaria ochroleuca* | 15.5 ± 0.28 [o] | 0.29 ± 0.05 [ghijk] |
| 31 | Water | | 0 [P] | 0 [m] |

Mean germination frequencies and mean radicle lengths are shown as ± Standard errors of means ($n = 15$). Same letters within each column indicate homogeneous groups between landraces according to the Tukey's HSD test ($p < 0.001$).

The radicle length of *S. hermonthica* treated with *Crotalaria* root exudates also differed significantly ($p < 0.0001$) among the landraces (Table 1). Between species, t-test revealed no significant differences in the radicle length with a *p* value of 0.1620 and t-value of −1.41138. The highest radicle length of 0.78 mm was recorded in the *C. ochroleuca* landrace from Kisumu coded as MKM204 while the lowest length (0.05 mm) was from a *C. ochroleuca* landrace from Busia coded as MB064. Four landraces of *C. ochroleuca* landraces from Kakamega, Homabay and Kisumu coded as MKKGO11, MKKG111, MHMY0215, and MKM0204 respectively had radicle lengths above 0.5 mm.

To determine if there was a relationship between germination frequency and radicle length, we carried out Pearson Correlations analysis. These results revealed a significant ($p < 0.0001$) positive correlation between *Striga* seed germination frequency and radicle length ($r = 0.371$).

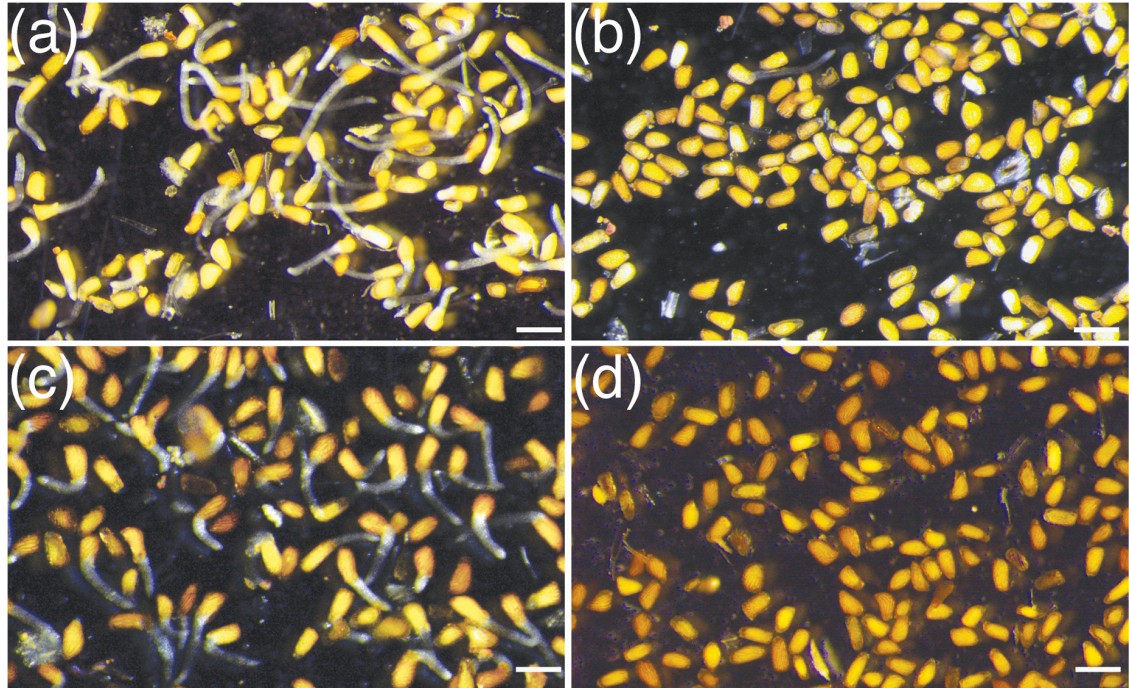

**Figure 1.** *Crotalaria* landraces displayed variation in their ability to stimulate *Striga* germination with regard to germination frequencies and radicle length of germinated parasite seedlings. (**a**) High induction of germination by *C. brevidens* landrace from Homabay County coded BHMBY0213. (**b**) Low induction of germination by *C. ochroleuca*, a landrace from Kisumu County coded MKSM0218 (**c**) Induction of germination by the synthetic strigolactone (GR24) – positive control and (**d**) Zero induction of germination by the negative control (water). Scale bar = 1 mm.

### 3.2. Crotalaria Blocks Striga Penetration at Multiple Levels up to the Pericycle

Successful *Striga* parasitism involve germination, attachment, penetration and establishment of a vascular connection [22]. This is well illustrated in the compatible interaction between *S. hermonthica* and the susceptible maize landrace Namba Nane where the parasite haustorium successfully penetrated host tissue subsequently establishing a vascular connection (Figure 2a). By 9 DAI, the parasite was well established and had developed vegetative tissue (Figure 2a). In contrast, *Striga-Crotalaria* interactions were characterized by incompatibilities at various levels. In the earliest stage of non-host incompatibility, illustrated in the landrace MKS218, *Striga* did not attach. Instead, the parasite got necrotic at the radicle and died (Figure 2b). In the second level (MKS204), parasite growth was blocked at the cortex (Figure 2c) while in the most persistent *Striga-Crotalaria* interactions (BMG0234), the parasite appeared to have reached the non-host's vascular cylinder (Figure 2d). However, similar to earlier incompatible interactions, parasite growth was slow and *Striga* did not burst its seed coat—a hallmark of unsuccessful vascular connection.

To gain further insights into the extent of parasite penetration in this interaction, we carried out more detailed histological analysis at the *Striga-Crotalaria* interphase and compared the observations to those of compatible *Striga*-maize interactions. *Striga*-maize interaction shows a well-developed haustorium, vascular tissue as well as the parasites food storage organ—hyaline body (Figure 3a). In contrast, parasite growth was inhibited in the *Striga-Crotalaria* interaction (Figure 3b). In this case, the parasite radicle succeeded in attaching on the *Crotalaria* root and formed a haustorium (Figure 3b). However, parasite ingression was blocked at the pericycle just before the entering the vascular cylinder (Figure 3b). The parasite was therefore unable to establish vascular connections with the non-host.

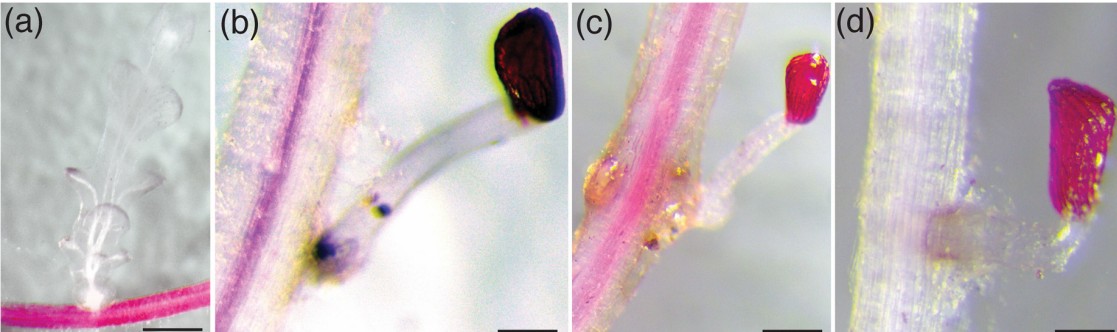

**Figure 2.** Levels of post germination interactions in compatible *Striga*-maize and incompatible *Striga-Crotalaria*. (**a**) Compatible *Striga*-maize interactions in a local maize landrace Namba Nane showing successful parasite penetration and a successful *Striga*-host xylem bridge. Soon after attachment and vascular connection, the parasite rapidly developed and became vegetative. Scale = 0.5 cm. (**b**–**d**) Different levels of *Striga-Crotalaria* incompatible interactions illustrated by (**b**) high incompatibility in the landrace MKS218 where the parasite was unable to attach; (**c**) where the parasite managed to penetrate *Crotalaria* but was blocked immediately after attachment (MKS204) and (**d**) furthest penetration of *Striga* into non-host tissue where the parasite appeared to approach the host vascular cylinder (BMG234). Scale = 1 mm. All interactions were determined 9 days after infection (DAI).

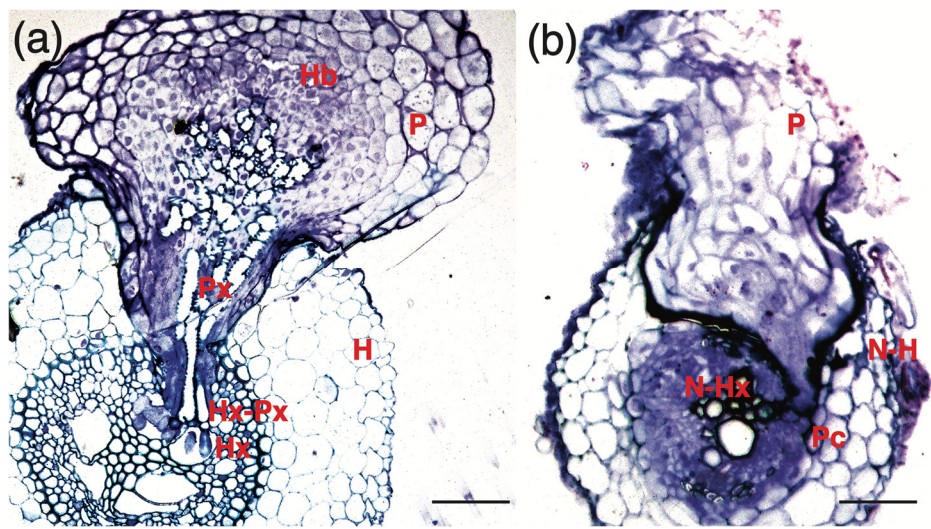

**Figure 3.** Post germination histological analysis of interactions between hosts and non-hosts. (**a**) Transverse section of compatible *Striga*-maize interaction showing a well-developed haustorium, establishment of *Striga*-host vascular connection (Px-Hx), parasite food storage organ – hyline body (Hb). Scale = 0.1 mm (**b**) Transverse section through unsuccessful parasitism in the landrace BMG234. The parasite is able to penetrate the non-host until it is inhibited at the pericle. As a result, the parasite is not able to establish vascular connections with the non-host and subsequently dies. All interactions were determined 9 DAI. Scale = 0.1 mm. H = Host; P = Parasite; Hx = Host Xylem; Px= Parasite Xylem; Px-Hx = Host Xylem connected to Parasite Xylem.

## 4. Discussion

Controlling *Striga* is difficult because the parasite produces a large number of seeds that remain viable for decades in soil, creating a large seed build up (seedbank). Depleting this seedbank requires stimulation of parasite germination without a host (suicidal germination). In this study, we determined effectiveness of different landraces of *Crotalaria* to stimulate germination of *Striga*. Germination ability was measured using percentage germination frequency as well as radicle length of the germinated *Striga* seedling. With regard to germination frequency, we found great variations within *Crotalaria* landraces.

Overall, the germination frequencies were comparable to previously tested hosts and non-hosts. For example, germination frequencies of between 8% and 66% were observed in soybean [*Glycine max* (L.) Merr.] [23] while in cotton (*Gossypium hirsutum* L. and *G. barbadense* L.), germination frequencies of between 13.3% and 50.0% were reported [24]. These differences are attributable to variations in production of different types and concentrations of strigolactones by the landraces. In general, 5-Deoxystrigol-like strigolactones have been found to be more potent in *Striga* germination [25] and stimulating hyphal branching [26]

We also used *Striga* radicle length to determine effectiveness of root exudates. We found that germination frequency generally correlated positively with radicle length implying that landraces with high germination frequencies also had long radicles. However, in the case of the landrace BHMBY0213, the germination frequency was high but radicle length was relatively small. Such differences can also be attributed to the composition and amounts of strigolactones in the roots of the plants. In this case, landraces that led to long *Striga* radicles can be assumed to stimulate germination of the seedling earlier that those landraces that lead to shorter radicles.

We further determined the level of non-host incompatibility in the *Striga-Crotalaria* interactions. The extent to which the parasite penetrates the host is important because even though the parasite dies, it may expose the host to other secondary infections. It is also possible that during the initial stages of parasitism, *Striga* could adversely affect a host or non-host through phytotoxity [15,16].

Yoshida and Shirasu, [14] described four levels of *S. hermonthica* non-host incompatibility as follows: Incompatibility expressed after vascular connection (level 1), endodermis and pericycle blockage level (II), mechanical barrier which occurs in root cortex level (III), and incompatibility preventing attachment level (IV) [14]. We observed blockage at levels IV (failure of attachment); III (blockage at the cortex) and in some instances, the parasite was able to penetrate the non-host past the cortex but got inhibited before it could enter the vascular cylinder (level II). Failure to attach is attributable to an ineffective haustorium while inhibition at levels III and IV can be due to biochemical or physiological non-host-parasite incompatibility [14]. However, the actual biological mechanisms underpinning these incompatibilities in *Striga-Crotalaria* interactions will require elucidation. Because incompatibility between *Striga* and *Crotalaria* occurs early during infection, such an interaction is unlikely to adversely affect the trap crop. Still, a possibility of such an effect should be considered in future investigations.

In summary, our study demonstrates use of laboratory-based germination and histological assays to determine the suitability of a *Striga* non-host as a trap crop. The assays provide valuable information on efficiency of germination stimulation and extent of parasite compatibility with the non-host. The assays are also applicable to other *Striga*-non-host interactions. Future studies will be necessary to determine if there is any macromolecular exchange between *Striga* and trap crops. The study also underscored the importance of *Crotalaria* as a germination stimulant for use in integrated *Striga* management practices because of its ability to stimulate *Striga* germination. The legume is consumed in the *Striga* prone region of western Kenya and therefore an attractive choice for technology uptake in *Striga* management practices. If adopted, *Crotalaria* will provide further benefits of increased food and nutrition security and improved soil fertility by nitrogen fixation in these smallholder farms.

**Supplementary Materials:** The following are available online at http://www.mdpi.com/2073-4395/10/6/873/s1, Table S1: Collection sites for twenty-nine Slenderleaf (*Crotalaria*) landraces used in the study.

**Author Contributions:** Conceptualization, S.R. and N.L.M.B.; methodology, F.A.M. and W.K.; formal analysis, F.A.M., J.O.N. C.O. and B.M.G.; resources, S.R. and N.L.M.B.; writing—original draft preparation, F.A.M., N.L.M.B., J.O.N. and S.R.; writing—review and editing, S.R., B.M.G., E.O.O. and P.K.K.; visualization, S.R., C.O. and W.K.; supervision, S.R., N.L.M.B., J.O.N., B.M.G.; project administration, N.L.M.B.; funding acquisition, S.R., N.L.M.B., E.O.O. and P.K.K. All authors have read and agreed to the submitted manuscript.

**Funding:** This research was funded by National Research Fund of Kenya.

**Acknowledgments:** We acknowledge help from greenhouse staff at Kenyatta University (Duncan Agwanda) in maintaining plant material.

**Conflicts of Interest:** The authors declare no conflict of interest. The funders had no role in the design of the study; in the collection, analyses, or interpretation of data; in the writing of the manuscript, or in the decision to publish the results.

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
