# Peer review of "Witchweed’s Suicidal Germination: Can Slenderleaf Help?"

_agronomy, doi:10.3390/agronomy10060873_

Round 1

Reviewer 1 Report

Date: 20/05/2020

Dear authors:

I have reviewed this manuscript and a report for you all is given below:

  1. Firstly, the manuscript should be written with double spaces all through.
  2. Title should include the Latin names of all the main organisms.
  3. The abstract emphasize heavily on the background of the study and a little weight on the methods, results. Therefore the conclusions drawn thereby in the abstract seems to be without much results.
  4. In the abstract and text, the authors should define any substandard term such as ‘landraces’.
  5. The introduction should include a clear hypothesis that the authors have tested and that reflects the tittle of the paper.
  6. Line 84-87, why results are in the introduction? This is very surprising!!!!
  7. Line 90, ‘thirteen’ should be written as ‘13’. As a standard rule, most journals would spell out numbers from 1 to 9 only.
  8. Line 93: this line is confusing, reword to clarify. Include crops or situation where the seeds were collected from.
  9. Line 95, seeds were pre-germinated? Explain briefly, how the seed lots were handled/ stored after collection and when (after what period of time) were the seed pre-germinated?
  10. Line 98, why to use the word plantlet instead of seedlings?
  11. Line 105, add one space between 500 and seed.
  12. Line 105-16, cite this statement.
  13. Line 108-10, cite this statement.
  14. Line 117, did you consider any specific length of radicle to be called as the seed has germinated?
  15. Line 141 what is Carnoy’s fixative? Please cite this even though it is quite a standard term!!!
  16. Line 45, ‘1hour’ should be ‘one hour’
  17. Line 149, how did you check for the infection?
  18. Line 153, 2 should be ‘two’
  19. Authors, you have not used the word ‘Slenderleaf’ n the ‘M & M’.
  20. Table 1, can you mention/describe/ refer to the treatments where the actual treatments coming from? Are you using the words/ terms: treatments, landraces, accession code, etc. alternatively? If so, why? Define clearly and use only one term for clarity and readability.
  21. Line 196-198, cite this statement.
  22. Line 272-277, any speculation for further studies? Or, how the method should be standardized so it can be performed quickly rather than going through whole lengthy processes?
  23. What is the implications for other scientists to use this method?
  24. Line 292, Reference #1 was not found in the text, unless I have really missed it!!!!

The Reviewer

Author Response

Thank you for a very detailed and critical review of our manuscript.

  1. Firstly, the manuscript should be written with double spaces all through.

We thank the reviewer for this observation. We adopted the formatting of the template provided but we will accept to reformat if this is not satisfactory

  1. Title should include the Latin names of all the main organisms.

We wanted to exploit multiple meanings of “suicide” and by the “slender leaf” in a pun on the title. We feel that the pun will be lost if we made it too crowded with the Latin names. In the abstract, the complete taxonomy of the organisms is provided.

  1. The abstract emphasize heavily on the background of the study and a little weight on the methods, results. Therefore the conclusions drawn thereby in the abstract seems to be without much results.

We agree. The abstract has now been revised to shorten the background and provide more details about the findings.

  1. In the abstract and text, the authors should define any substandard term such as ‘landraces’.

Landrace has been defined in lines 26 and 84.

  1. The introduction should include a clear hypothesis that the authors have tested and that reflects the tittle of the paper.

We have revised part of the introduction: Lines 83-86.

  1. Line 83-86, why results are in the introduction? This is very surprising!!!!

We thank the reviewer for this observation. Results have been expunged from the introduction.

  1. Line 90, ‘thirteen’ should be written as ‘13’. As a standard rule, most journals would spell out numbers from 1 to 9 only.

This has been revised: Line 89.

  1. Line 93: this line is confusing, reword to clarify. Include crops or situation where the seeds were collected from.

This has been clarified. The Crotalaria seed collections are propagated and maintained by farmers. Striga seeds were collected from sorghum infested fields in Western Kenya region. Lines 89-96.

  1. Line 95, seeds were pre-germinated? Explain briefly, how the seed lots were handled/ stored after collection and when (after what period of time) were the seed pre-germinated?

This has been clarified: Lines 89-96.

  1. Line 98, why to use the word plantlet instead of seedlings?

This has been revised: Line 102.

  1. Line 105, add one space between 500 and seed.

Revised: Line 110

  1. Line 105-16, cite this statement.

Cited line 113

  1. Line 108-10, cite this statement.

Cited line 115

  1. Line 117, did you consider any specific length of radicle to be called as the seed has germinated?

No. Seeds were considered to have germinated when a white radicle protruded through the seed coat.

  1. Line 141 what is Carnoy’s fixative? Please cite this even though it is quite a standard term!!!

Citation for Cornay’s fixative has been provided. Line 154

  1. Line 45, ‘1hour’ should be ‘one hour’

Revised: Line 158

  1. Line 149, how did you check for the infection?

This was carried out after section and mounting on microscope slides. Slides were observed, and photographed using a Leica DM100 microscope (Leica, Germany) fitted with a Leica MC190HD camera (Leica, Germany). Lines 168-170

  1. Line 153, 2 should be ‘two’

Revised: line 167

  1. Authors, you have not used the word ‘Slenderleaf’ n the ‘M & M’.

We have included the word “slenderleaf in materials and methods”

  1. Table 1, can you mention/describe/ refer to the treatments where the actual treatments coming from? Are you using the words/ terms: treatments, landraces, accession code, etc. alternatively? If so, why? Define clearly and use only one term for clarity and readability.

We thank the reviewer for this observation. We have adopted use of the word “landrace”

  1. Line 196-198, cite this statement.

We have provided a citation.

  1. Line 272-277, any speculation for further studies? Or, how the method should be standardized so it can be performed quickly rather than going through whole lengthy processes?

The conclusion statement has been revised to provide implications of our work as well as prospects for future research. Lines 290-300.

  1. What is the implications for other scientists to use this method?

The conclusion statement has been revised to provide implications of our work as well as prospects for future research.

This has not been revised: Lines 290-300

  1. Line 292, Reference #1 was not found in the text, unless I have really missed it!!!!

Thank you. The reference is now cited: Lines 40 and 44.

Reviewer 2 Report

The manuscript “Witchweed’s Suicidal Germination: Can Slenderleaf Help?” constitutes an interesting study concerning the importance of legumes (Crotalaria) as a germination stimulant for use in integrated Striga management practices. The manuscript is well described, especially the Materials and methods section, and the experimental design is appropriate. However, the results should be presented more clearly and the authors should improve statistical analysis and conclusions.

Lines 84-87: These sentences these sentences should be moved to conclusions. On the contrary, the aims of the study should be clearly spelled out.

Line 122: Since two landraces have been considered for Crotalaria, the analysis of variance should first check whether there are significant differences between the two landraces. Therefore, it is necessary to introduce the landraces factor in the statistical analysis. Please, modify the analysis, also in results section.

Table 1: Please, insert the number of samples as indicated in table S1. It would be easier to read comments if the sample number rather than the accession code were used in the text.

Table 1: The germination frequency should only have one decimal place and not two. Please, modify means and standard deviations in table 1 and text (lines 161, 163, 165, 168, 171).

Table 1: Please, verify significance of MKS0221 sample for germination frequency.

Since the authors cited other factors that can influence the germination ability of the trap crop in the rhizosphere (lines 251-254), such as soil microbiome [24] soil texture, or the soils phosphate nutrition content, the authors have to describe these information in the paper and relate them with results. In particular, these factors can contribute to explain differences between values of the same landrace in table 1. To do this, these factors, if available, should be inserted into a GLM analysis.

Line 272-277: Conclusions should be improved, following what the authors wrote in the lines 72-75.

Author Response

  1. The manuscript “Witchweed’s Suicidal Germination: Can Slenderleaf Help?” constitutes an interesting study concerning the importance of legumes (Crotalaria) as a germination stimulant for use in integrated Striga management practices. The manuscript is well described, especially the Materials and methods section, and the experimental design is appropriate. However, the results should be presented more clearly and the authors should improve statistical analysis and conclusions.

We thank you for your positive comments on our manuscript. Statistical analyses and conclusions have been revised as per your suggestions. Lines 175 and 178

  1. Lines 84-87: These sentences these sentences should be moved to conclusions. On the contrary, the aims of the study should be clearly spelled out.

The last paragraph in the introduction has been replaced with a revised one that describe the study hypothesis. Lines 83-86

  1. Line 122: Since two landraces have been considered for Crotalaria, the analysis of variance should first check whether there are significant differences between the two landraces. Therefore, it is necessary to introduce the landraces factor in the statistical analysis. Please, modify the analysis, also in results section.

We have done analysis for significant differences between the Crotalaria species. There were no significant differences. Revisions are reported in the materials and methods as well as the results sections.

  1. Table 1: Please, insert the number of samples as indicated in table S1. It would be easier to read comments if the sample number rather than the accession code were used in the text.

Table 1 has been revised

  1. Table 1: The germination frequency should only have one decimal place and not two. Please, modify means and standard deviations in table 1 and text (lines 161, 163, 165, 168, 171).

These have been revised

  1. Table 1: Please, verify significance of MKS0221 sample for germination frequency.

This has been revised

  1. Since the authors cited other factors that can influence the germination ability of the trap crop in the rhizosphere (lines 251-254), such as soil microbiome [24] soil texture, or the soils phosphate nutrition content, the authors have to describe these information in the paper and relate them with results. In particular, these factors can contribute to explain differences between values of the same landrace in table 1. To do this, these factors, if available, should be inserted into a GLM analysis.

As we did not conduct field analysis, we have expunged this section in the discussion.

  1. Line 272-277: Conclusions should be improved, following what the authors wrote in the lines 72-75.

Conclusions have been revised and aligned with the study hypotheses outlined in the introduction. Lines 290-300.

Reviewer 3 Report

This paper describes laboratory studies to investigate the effectiveness of different Croalaria accessions at inducing germination of striga, as well as microscopic analysis of post-germination interactions. The paper is generally well written. The finding of quite different levels of germination stimulation (from 16 to 55%) sets up some interesting follow-up work, in which the most and least effective accessions in the lab could be evaluated under field conditions to determine how they compare at reducing the striga seed bank. As is, the paper is interesting, but provides a relatively small incremental advance. From a technical standpoint, my primary concern is that it appears the experiments were not repeated in time. For example, the data in table 1 apparently come from three replications of each accession in a single experiment. Without a separate run it is difficult to be confident that the observed differences among accessions are real and repeatable. Minor comments/suggestions follow.

Ln 42 replace “genera” with “species”

Ln 113 correct or provide more information for the source of GR24. I tried to google this, but could not find it

Ln 120  What is meant by “a smooth curve”?

Ln 127-128 I assume you mean the landrace that induced the lowest frequency of striga germination (not the landrace that had the lowest germination). Why chose this one? The landrace that induced the highest frequency of striga germination might be the most compatible one, and therefore might show greater post-germination interaction.

Ln 136  What is meant by “per genotype?” It seems that there was only one genotype each of maize, Crotalaria, and striga.

Ln247-249  These two sentences are not justified by the data, since no field studies were done to confirm the laboratory assays. Production of germination-stimulation compounds might vary under field conditions. The authors provide these caveats in the same paragraph, so do not lead off with such definitive statements.

Author Response

  1. We thank you for positive comments on our manuscript. Your primary concern was with regard to repeating the experiments for more statistical confidence.

Because experiments were done under controlled conditions with minimal environmental conditions; procedures and design, we did not expect to find significant variations with repeated experiments.

  1. Ln 42 replace “genera” with “species”

Revised: line 41

  1. Ln 113 correct or provide more information for the source of GR24. I tried to google this, but could not find it

A link is provided for GR24: Line 118

  1. Ln 120  What is meant by “a smooth curve”?

It is a feature of imageJ used to measure distances of objects that are not linear. This has been clarified: Lines 127

  1. Ln 127-128 I assume you mean the landrace that induced the lowest frequency of striga germination (not the landrace that had the lowest germination). Why chose this one? The landrace that induced the highest frequency of striga germination might be the most compatible one, and therefore might show greater post-germination interaction.

Yes, we mean the landrace showing the lowest Striga germination frequency. You raise an important point. Comparisons of compatibilities between landraces and Striga will require detailed analysis of frequencies of attachments, haustoria formation and vascular connection. We screened medium as well as high germination inducers and then selected the landrace that was most compatible. This was also the landrace induction of germination. We are requesting to revise figure 2 and add an additional figure to make this point clear.

  1. Ln 136  What is meant by “per genotype?” It seems that there was only one genotype each of maize, Crotalaria, and striga.

The statement has been revised.

  1. Ln247-249  These two sentences are not justified by the data, since no field studies were done to confirm the laboratory assays. Production of germination-stimulation compounds might vary under field conditions. The authors provide these caveats in the same paragraph, so do not lead off with such definitive statements.

Thank you for this important suggestion we have removed the part of the discussion that describe field experiments.

Round 2

Reviewer 2 Report

The authors should still make the following changes, two of which have previously been reported:

Line 163: The word "determine" is repeated twice

Table 1 : The germination frequency should only have one decimal place and not two. Please, modify standard deviations in table 1

Table 1: Please, verify significance of MKS0221 sample for germination frequency. It is not possible that significant group for MKS0221 is “cde” but only “de”.

Author Response

Line 163: The word "determine" is repeated twice

Thank You. We have revised.

Table 1 : The germination frequency should only have one decimal place and not two. Please, modify standard deviations in table 1

Thank you. This has been revised.

Table 1: Please, verify significance of MKS0221 sample for germination frequency. It is not possible that significant group for MKS0221 is “cde” but only “de”.

We have checked the significances of MKS0221 and revised accordingly

Reviewer 3 Report

I appreciate that the authors have pared down the discussion section, limiting the degree to which their laboratory results can be directly transferred to the field.

Author Response

Thank you.